# Genotyping of Circulating Free DNA Enables Monitoring of Tumor Dynamics in Synovial Sarcomas

**DOI:** 10.3390/cancers14092078

**Published:** 2022-04-21

**Authors:** Anja E. Eisenhardt, Zacharias Brugger, Ute Lausch, Jurij Kiefer, Johannes Zeller, Alexander Runkel, Adrian Schmid, Peter Bronsert, Julius Wehrle, Andreas Leithner, Bernadette Liegl-Atzwanger, Riccardo E. Giunta, Steffen U. Eisenhardt, David Braig

**Affiliations:** 1Department of Plastic and Hand Surgery, Medical Center - University of Freiburg, Faculty of Medicine, University of Freiburg, 79106 Freiburg, Germany; anja.eisenhardt@uniklinik-freiburg.de (A.E.E.); zacharias.brugger@uniklinik-freiburg.de (Z.B.); ute.lausch@uniklinik-freiburg.de (U.L.); jurij.kiefer@uniklinik-freiburg.de (J.K.); johannes.zeller@uniklinik-freiburg.de (J.Z.); alexander.runkel@uniklinik-freiburg.de (A.R.); adrian.schmid@uniklinik-freiburg.de (A.S.); steffen.eisenhardt@uniklinik-freiburg.de (S.U.E.); 2Institute for Surgical Pathology, Medical Center - University of Freiburg, Faculty of Medicine, University of Freiburg, 79106 Freiburg, Germany; peter.bronsert@uniklinik-freiburg.de; 3Tumorbank Comprehensive Cancer Center Freiburg, Medical Center - University of Freiburg, Faculty of Medicine, University of Freiburg, 79106 Freiburg, Germany; 4Department of Medicine I, Medical Center - University of Freiburg, Faculty of Medicine, University of Freiburg, 79106 Freiburg, Germany; julius.wehrle@uniklinik-freiburg.de; 5Department of Orthopedics and Trauma, Medical University of Graz, 8036 Graz, Austria; andreas.leithner@medunigraz.at; 6Diagnostic and Research Institute of Pathology, Medical University of Graz, 8010 Graz, Austria; bernadette.liegl-atzwanger@medunigraz.at; 7Division of Hand, Plastic and Aesthetic Surgery, University Hospital, Ludwig Maximilian University of Munich, 80336 Munich, Germany; r.giunta@med.uni-muenchen.de

**Keywords:** soft tissue sarcoma, synovial sarcoma, next-generation sequencing, targeted sequencing, circulating tumor DNA, ctDNA, liquid biopsy, diagnostic biomarker

## Abstract

**Simple Summary:**

Synovial sarcomas (SS) are rare soft tissue tumors of mesenchymal origin. Following resection of the primary tumor, about one third to half of the patients suffer from recurrence. Detection of local and distant recurrence during follow-up is commonly accomplished by imaging. There are no biomarkers available for routine diagnostics. We employ a highly sensitive targeted next-generation sequencing approach to monitor tumor dynamics by genotyping of circulating free DNA (cfDNA) in SS patients. cfDNA which harbors tumor-specific mutations (circulating tumor-DNA; ctDNA) correlated with the presence of viable tumor tissue. This enables timely and non-invasive detection of tumor recurrence and monitoring of treatment response independent of the anatomic location.

**Abstract:**

Background: Synovial sarcoma (SS) is a malignant soft tissue tumor of mesenchymal origin that frequently occurs in young adults. Translocation of the SYT gene on chromosome 18 to the SSX genes on chromosome X leads to the formation of oncogenic fusion genes, which lead to initiation and proliferation of tumor cells. The detection and quantification of circulating tumor DNA (ctDNA) can serve as a non-invasive method for diagnostics of local or distant tumor recurrence, which could improve survival rates due to early detection. Methods: We developed a subtype-specific targeted next-generation sequencing (NGS) approach specifically targeting SS t(X;18)(p11;q11), which fuses *SS18* (*SYT*) in chromosome 18 to *SSX1* or *SSX2* in chromosome x, and recurrent point mutations. In addition, patient-specific panels were designed from tumor exome sequencing. Both approaches were used to quantify ctDNA in patients’ plasma. Results: The subtype-specific assay allowed detection of somatic mutations from 25/25 tumors with a mean of 1.68 targetable mutations. The minimal limit of detection was determined at a variant allele frequency of 0.05%. Analysis of 29 plasma samples from 15 tumor patients identified breakpoint ctDNA in 6 patients (sensitivity: 40%, specificity 100%). The addition of more mutations further increased assay sensitivity. Quantification of ctDNA in plasma samples (*n* = 11) from one patient collected over 3 years, with a patient-specific panel based on tumor exome sequencing, correlated with the clinical course, response to treatment and tumor volume. Conclusions: Targeted NGS allows for highly sensitive tumor profiling and non-invasive detection of ctDNA in SS patients, enabling non-invasive monitoring of tumor dynamics.

## 1. Introduction

Compared to other soft tissue sarcomas, synovial sarcoma (SS) occurs mostly in younger adults with a median age diagnosis of 35 years [1]. Standard treatment for SS is multimodal, consisting of tumor resection, radiotherapy and/or chemotherapy. Five-year overall survival for tumors without metastases reaches approximately 90% in the pediatric population, whereas it is only 13% for those with metastases [2,3,4]. Genetic profiling is currently a standard clinical practice to investigate and monitor tumor behavior. SS is characterized by the chromosomal translocation t(X;18)(p11;q11), which fuses *SS18* (*SYT*) in chromosome 18 and *SSX1*, *SSX2* or *SSX4* in chromosome X [5,6]. The translocation is unique and is considered the main oncogenic driver in SS [7,8]. Additional point mutations with a frequency of >5% have been described in SS in several tumor-promoting genes, e.g., *TP53* [9].

Despite its significance as a diagnostic marker, the clinical targeting of this translocation remains elusive [10]. In a case report, the detection of the mRNA of the *SYT-SSX* fusion genes from circulating tumor cells was accomplished [11]. Our group could show that the mRNA of these fusion genes is delivered packaged in microvesicles from sarcoma cells [12].

Free circulating tumor DNA (ctDNA) in the peripheral blood of patients could serve as a non-invasive diagnostic biomarker to determine disease activity. In this regard, we have already succeeded in determining tumor-specific translocations and point mutations in myxoid liposarcoma (MLS) patients. We previously developed a highly sensitive targeted next-generation sequencing (NGS) approach specifically targeting MLS translocations t(12;16) and t(12;22) and recurrent point mutations [13]. In this study, we adopt this approach to quantify ctDNA in SS.

## 2. Material and Methods

### 2.1. Study Population

Samples in this study were obtained from 25 SS patients in total. The cohort contained 15 patients who were treated at the Comprehensive Cancer Center, Freiburg (CCCF, Freiburg, Germany). An independent cohort consisted of 10 tumor samples obtained from the Diagnostic and Research Institute of Pathology, Medical University of Graz (Graz, Austria) in cooperation with the Biobank of the Medical University of Graz. Formalin-fixed paraffin-embedded (FFPE) tumor tissue, plasma and whole blood samples were available for analysis.

### 2.2. Blood Sampling

All blood samples were collected and processed as described previously [14].

### 2.3. Cell Lines

HS-SY-II and FUJI cell lines were established by H. Sonobe (Kochi Medical University, Kochi, Japan) and T. Nojima (Kanazawa Medical University, Kanazawa, Japan), respectively. HS-SY-II cells genetically occur as an SYT-SSX1 fusion transcript, whereas FUJI cells possess SYT-SSX2.

HS-SY-II cells were maintained in DMEM supplemented with 10% fetal bovine serum and 1% penicillin-streptomycin (PS) [15] and FUJI was maintained in RPMI 1640 with 10% fetal bovine serum and 1% PS [16].

### 2.4. Isolation of DNA from FFPE Tumor Tissue

DNA was extracted from FFPE tissue according to the manufacturer’s instructions, using the FFPE Qiagen Extraction Kit (Qiagen, Hilden, Germany). Approximately 8 sections of 5–10 µm were digested with Proteinase K at 56 °C for 3 days. DNA was eluted in 80 µL of RNase-free water.

### 2.5. Isolation of DNA from Blood/Leukocytes

DNA was extracted with the DNeasy Blood and Tissue Kit (Qiagen, Hilden, Germany). The DNA was eluted in a volume of 200 µL RNase-free water.

### 2.6. Cell-Free DNA Isolation

Cell-free DNA was extracted as described previously [14]. A total of 1 mL to 5 mL of plasma was used and cfDNA was eluted in 30 µL of AVE buffer.

### 2.7. Quantification of cfDNA

Quantity of cfDNA was determined by Qubit 3.0 Fluorometer, using a high-sensitivity Qubit dsDNA HS Assay Kit (Invitrogen, Carlsbad, CA, USA).

### 2.8. Library Preparation for Tumor, Leukocyte and cfDNA Samples and Next-Generation Sequencing

DNA from tumor tissue and matched normal controls from leukocytes were prepared for sequencing by generating libraries using NEB NEBNext Ultra II FS DNA Library Prep Kit for Illumina (New England Biolabs, Ipswich, MA, USA). Tumor and leukocyte DNA were fragmented to optimal length (approximately 150 bp) by incubation with 1 µL NEBNext Ultra II FS Enzyme for 25 min at 37 °C. A total of 60 ng DNA input was used, followed by 8 polymerase chain reaction (PCR) cycles.

Libraries from cfDNA were generated according to the manufacturer’s manual Takara SMARter Thruplex Tag Seq 48S Kit (Takara Bio Inc., Kusatsu, Shiga, Japan). A total of 10 ng input of cfDNA was used and adapters were ligated to attach unique molecular identifiers (UMI) to each side of the DNA fragment with single indices.

### 2.9. Panel Design and Hybridization

Firstly, a subtype-specific (i.e., SS) panel was designed (Appendix A) which encompassed breakpoint regions and exons of genes that are hotspot regions in SS [17,18,19,20].

Secondly, a breakpoint lockdown panel was designed, which only encompassed the specific breakpoint regions detected in the SS patients. These breakpoints were identified by sequencing tumor tissue with the subtype-specific panel.

Thirdly, a patient-specific exome panel was designed based on the mutations identified in exome sequencing (CeGaT GmbH, Tübingen, Germany) performed on FFPE and leukocyte tumor DNA from patient 6. Panel sizes are depicted in Appendix A.

The IDT xGen panels comprising biotin probes were used to hybridize target regions. DNA concentration of the libraries was measured with Qubit (Invitrogen, Carlsbad, CA, USA) and equal amounts of each library were pooled (between 100 and 250 ng per library). Target regions were pulled down with streptavidin-coated magnetic beads. The hybridization protocol was performed according to the manufacturer’s instructions with xGen Hybridization and Wash kit (Integrated DNA Technologies, Coralville, IA, USA). Following hybridization and PCR, 16 and 9 cycles, post-capture samples were size selected using AMPure XP beads (Beckman Coulter, Brea, CA, USA). Double capture was performed with libraries of 100–250 ng pooled to total 500 ng—1 µg. Both captures were incubated for 4 h at 65 °C.

### 2.10. Sequencing

The length of the cleaned-up captured libraries was then measured by Tape Station Agilent D500 (Agilent, Santa Clara, CA, USA). For accurate DNA concentration, libraries were measured with qPCR Light Cycler 480 System (Roche Diagnostics, Basel, Switzerland), using the NEBNext Library Quant Kit for Illumina (New England Biolabs, Ipswich, MA, USA).

The desired amount of the libraries was then calculated and sequenced using a MiSeq system with paired end reads, MiSeq V2 300 cycle (Illumina Inc., San Diego, CA, USA).

### 2.11. Bioinformatics

The sequence reads of tumor tissue samples and controls (tumor native or FFPE) were uploaded to Illumina BaseSpace (Illumina Inc., San Diego, CA, USA) and aligned to human genome (hg38). Marking for duplicates and calling somatic mutations including single nucleotide variants and short indels were detected using DRAGEN Somatic Pipeline (BaseSpace, Illumina Inc., San Diego, CA, USA). Patient-matched blood samples were analyzed simultaneously in order to eliminate germline variants. For structural variant analysis, the sequence reads were uploaded in Integrated Genomics Viewer (IGV), aligned to *Homo sapiens* (human) genome assembly GRCh38 (hg38). The reads were ordered by chromosome of mate and the exact positions of the corresponding read on chromosome X were combined. The resulting breakpoints were verified by creating an individual custom FASTA sequence for each patient and by realigning the sequence reads.

The sequence reads of cfDNA samples containing UMIs were uploaded to the Curio Genomics web platform (https://curiogenomics.com, (accessed on 13 January 2020)) and aligned to GRChg38 using Bowtie2. UMI-family consensus reads were called and somatic mutations including single nucleotide variants were discovered with the Curio Genomics built-in analysis pipeline. Structural variants in cfDNA samples containing UMIs were detected using Galaxy (www.usegalaxy.eu, accessed on 20 January 2020)). In this process, UMIs were extracted from the sequence reads and subsequently, consensus reads were assembled and aligned to the previously described custom FASTA sequences. To reduce the rate of false positives, each alteration was manually reassessed with the Integrative Genomics Viewer (IGV) [21]. In order to obtain general information about the performance of the panels, the sequence reads were further processed with Illumina DRAGEN Enrichment Pipeline (BaseSpace, Illumina Inc., San Diego, CA, USA). All events were documented in GRCh38 assembly.

### 2.12. Tumor Volume Rendering

Tumor volume was calculated as described previously [14]. In brief, image data from MRI and computer tomography (CT) acquisitions were used for tumor volume analysis. The tumors and metastasis were delineated against normal tissue and volume rendering was calculated using HOROS (Horosproject.org, accessed on 5 July 2021).

## 3. Results

### 3.1. Detection of Breakpoints and Point Mutations in SS Tumors by Targeted NGS

We designed a subtype-specific tumor enrichment panel for detection of breakpoints and point mutations with a reported frequency of at least 5% in SS [17,18,19,20]. The 43,536 bp panel covers the introns of *SYT*, *SSX1* and *SSX2*, where the translocation occurs, *E-Cadherin*, *EGFR*, *TP53* and other mutation hotspots within exons from nine genes in total (Appendix A). We evaluated the performance of the panel for tumor FFPE tissue. The cohort consisted of 25 SS tumors of which 15 had leukocyte DNA as matched-normal control and two cell lines, FUJI and HS-SY-II. At least one breakpoint could be identified in all samples and both breakpoints could be identified in 17 out of 25 tumors and in the FUJI cell line. No point mutations were identified in the tumor samples with matched-normal controls, and no false positive events (breakpoints or point mutations) were identified in the matched-normal controls (Figure 1A). In summary, a mean of 1.68 mutations was identified in tumor tissue, which can be tracked in cfDNA. The sensitivity and specificity for detection of at least one breakpoint was 100%. For simultaneous detection of both breakpoints of the reciprocal translocation, the sensitivity was 68% and specificity 100%. We could determine loss and gain of DNA at the breakpoint region for the 17 tumors where we identified both breakpoints (Figure 1B). A mean gain of 144 bp (SD 526 bp) occurred next to the break of SYT on chromosome 18 and loss of 32 bp (SD 146 bp) on chromosome X. Patient 19 had the largest gain (1551 bp); the highest loss occurred in patient 14 (642 bp). The mean coverage of each tumor sample was 889 x (SD 357 x), and the mean on-target rate after double-capture was 86.5% (SD 6.2%) (Figure 1C).

### 3.2. Detection of ctDNA in SS Patients by Breakpoint Sequencing

In order to determine the sensitivity and specificity of our technique for ctDNA detection, a breakpoint panel that included the breakpoint region of 15 SS patients (patient 1 to patient 15) was designed. Every patient-specific breakpoint was covered with a 360 bp stretch of three 120 bp lockdown probes, 180 bp on chromosome X and 180 bp on chromosome 18. In total, 29 plasma samples of these 15 patients were analyzed; 8 of the 15 patients had “active disease” (primary tumor (*n* = 2) or metastatic disease (*n* = 6)), 7 were in complete remission. In total, 15 plasma samples from these 8 “active” patients were sequenced and analyzed. Breakpoint ctDNA was identified in 40% of these samples (Figure 2A). In 1 of the 2 patients with primary tumors, and 5 of the 13 samples with metastatic disease, breakpoint ctDNA was identified (Figure 2A). Relative quantification of ctDNA revealed that 0% to 26.2% of cfDNA was tumor derived, with a mean of 2% (Figure 2B). Absolute quantification showed that ctDNA was detectable from 0 reads per ml to 1560 reads per ml with a mean of 111 reads per ml (Figure 2C). Comparing ctDNA concentrations to tumor volumes, as determined by imaging, revealed that larger tumors tend to shed more ctDNA (Figure 2D). Calculations showed a correlation coefficient of 0.55 with a *p* value of 0.0276. In total, 7 of the 15 patients were in remission, of which 14 plasma samples were taken. No false-positive breakpoint reads were identified. The sensitivity of the assay in our cohort was 40% and the specificity 100%.

### 3.3. Assay Sensitivity Can Be Increased by Incorporation of Point Mutations in the Panel

To enable detection of minute amounts of SS tumor DNA with extreme accuracy, we investigated whether addition of point mutations to the panel increases sensitivity. A total of 10 ng of cfDNA from a healthy donor was spiked with decreasing amounts of FUJI cell line DNA, resulting in a variant allele frequency (VAF) of 2.5%, 0.25%, 0.05% and 0%. DNA fragments were individually tagged with a unique molecular identifier (UMI) at the 5′ and 3′ end. In addition to one breakpoint, single nucleotide variants (SNVs) were identified in PIK3CA and APC (exon 11, 13 and 15). This allowed us to track five mutations simultaneously (one breakpoint, four SNVs). Tumor DNA could still be detected at a VAF of 0.05%. There was only one false positive read in the 0% VAF sample (Figure 3). Performance data on the dilution series are detailed in Appendix A.

### 3.4. Detection of ctDNA in an SS Patient with Patient-Specific Panel

Tumor tissue and matched leukocyte DNA from patient 6 were subjected to exome sequencing to identify additional targetable tumor mutations. These mutations were incorporated in a patient-specific exome panel in addition to the breakpoints identified by the standard panel, resulting in a panel size of 6291 bp (Appendix A). After panel validation with tumor tissue and matched leukocyte DNA, this allowed targeting of 18 genomic regions in cfDNA.

The patient presented with a 38 cm^3^ large SS in his right lower leg. Over three years, the patient visited our department for treatment and routine follow-ups. During this time, we collected 11 plasma samples for ctDNA quantification. The patient initially received neoadjuvant radiotherapy and the tumor was subsequently completely resected (Figure 4A). We observed a decline in ctDNA concentration following resection (sample 1–3), from 0.4% (S1) to 0.07% (S2). The patient continued with routine follow-ups over the next three years. A few months after the primary tumor resection, we already observed an increase in ctDNA levels without any evidence of tumor recurrence in CT or MRI imaging (sample 4 + 5). About one year later, a calvarial metastasis was discovered and resected (44 cm^3^). ctDNA levels mirrored the clinical course with high ctDNA concentrations before the resection and a drastic decrease postoperatively (sample 6 + 7). ctDNA levels again increased upon the detection of bilateral lung metastases, and decreased after incomplete resection (sample 8 + 9). More lung metastases were resected in a second operation followed by adjuvant chemotherapy (sample 10 + 11) (Figure 4A). The breakpoint panel detected ctDNA only in sample 6; therefore, its performance was inferior to the exome panel. Mutations in STAP-2, KIAA1328 and MRPS16 were most commonly detected in ctDNA. Each mutation allowed individual tracking of tumor dynamics in this patient (Figure 4B). ctDNA correlated with the clinical course and detected recurrence about one year prior to routine follow-up examinations according to current guidelines.

## 4. Discussion

In localized SS disease, surgical resection combined with multimodal therapy enables a curative approach. However, approximately one third to fifty percent of all patients experience a local recurrence or metastases after the primary resection [4]. Although it remains a matter of debate whether surveillance imaging improves survival, current guidelines recommend follow-up with various imaging modalities [22,23]. Local recurrences are usually detected by MRI or ultrasound, pulmonary metastases with conventional chest radiographs or CT scans. Surveillance of other anatomical sites is usually not necessary, as metastases to extrapulmonary sites are an exception in SS. This is different from other sarcoma subtypes, which frequently show distant spread outside the lungs [24]. New examination methods, which enable earlier detection of recurrence, might improve tumor monitoring during follow-up.

ctDNA is a potential biomarker of recurrence in patients after tumor resection and allows monitoring of tumor activity and treatment response in metastatic sarcoma patients [13,14,25]. As ctDNA has a short half-life (approximately 2 h) it can allow an accurate snapshot of the genomic landscape of the tumor [26]. Although ctDNA may surpass the sensitivity of current imaging modalities, it remains elusive as to whether this could translate into a survival benefit for patients suffering from tumor relapse.

As each SS tumor harbors a unique breakpoint sequence within *SYT* and one of three *SSX* genes (*SSX1*, *SSX2* and *SSX4*) forming the tumor-specific translocation t(x;18)(p11.2;q11.2), these breakpoint sequences can be used for ctDNA detection. We designed a subtype-specific panel that covered the breakpoint regions in order to sequence and validate each individual breakpoint. As each breakpoint has a unique sequence, this allows tracking of each tumor individually with high specificity. False positive results from tracking of point mutations in common cancer genes, which might arise from age-related clonal hematopoiesis, therefore do not affect the specificity of our assay [27]. For breakpoint detection in tumor FFPE tissue, we observed an unprecedented sensitivity of 100% and specificity of 100% in matched-normal controls.

Previously, we presented a methodological and clinically feasible approach for ctDNA monitoring in translocation-associated myxoid liposarcoma (MLS) patients in a routine diagnostic setting using a disease-specific hybrid capture NGS technique. In this study, we adapted the diagnostic assay to the genetic landscape of SS in order to non-invasively monitor patients. The assay performance for detection of breakpoints was similar, with a mean of 1.68 breakpoints per tumor. However, as point mutations occurred less frequently in the SS cohort, the overall number of trackable mutations is lower in SS than in MLS [13]. Still, NGS allowed the detection of mutations with a VAF of 0.05% with breakpoint detection only (Figure 3). Screening plasma samples of SS patients with this assay revealed high specificity of 100%, but limited sensitivity of 40%, despite the assay’s 0.05% VAF limit of detection (Figure 2). This is most likely due to very low ctDNA concentrations in SS patients, as seen in other sarcoma subtypes. However, by addition of point mutations obtained from exome sequencing of individual tumors, assay sensitivity can be further increased [13]. Focusing on a patient with a localized tumor of the lower extremities, who later developed metastatic disease, we established this patient-specific approach for SS with ultra-high sensitivity. Point mutations were identified by tumor exome sequencing and breakpoints from panel sequencing. Mutations which are described as pathogenic, e.g., in COSMIC (https://cancer.sanger.ac.uk/cosmic, (accessed on 16 October 2019)), were included in the patient’s individual exome panel together with the mutations identified in the subtype-specific panel, resulting in 18 targetable mutations. Among these were *ACOT11*, *MRPS16*, *P2RX7*, *KIAA1328* and *STAP-2*. This improved assay enables real-time monitoring of tumor activity even in localized, small tumors, independent of their anatomic location.

ctDNA in serial blood samples collected during the treatment of this patient was correlated with the clinical course and tumor burden. The exome panel was more effective and sensitive than the breakpoint panel (Figure 4A). Even the small primary tumor was detected, and following resection, ctDNA decreased. The patient subsequently developed several metastases, one calvarial, as well as bilateral lung lesions. The ctDNA increased for approximately a year before the calvarial metastasis was identified during routine follow-up. This example demonstrates a major advantage of liquid biopsy over standard imaging. Tumor detection occurs independently of the anatomic site, which is especially important for detection of extra-pulmonary metastases. In this scenario, the rise of ctDNA shortly after tumor resection would have entailed more extensive imaging and thus earlier detection and treatment of the calvarial metastasis.

Comprehensive mutation profiling with an SS-specific panel and additional exome sequencing comes at some cost. Still, the assay can be incorporated into a feasible workflow in a real-world setting. Sequencing costs are constantly decreasing and exome sequencing is increasingly performed in routine treatment of sarcoma patients. We are currently improving our bioinformatic pipeline, so that evaluation of tumor-specific mutations and the generation of a patient-specific panel, as well as quantification of tumor mutations in cfDNA, are fully automated. Together with a standardized workflow that runs largely automated on pipetting machines in a diagnostic laboratory, this will enable cost- and time-effective analysis.

Besides the inclusion of additional mutations from exome sequencing, the sensitivity of the test can be improved by other variables. This is important in translocation-associated sarcomas, as they harbor only a few additional mutations besides the driver translocations. One possibility is to enhance the conversion rate of the assay. This term describes the relative amount of cfDNA molecules which are converted to the library, sequenced and finally analyzed. We therefore employed library preparation kits that are optimized for challenging cfDNA samples and performed a double target capture of the libraries [28]. Another possibility is to increase the volume of plasma for cfDNA extraction. In total, 1 ml of plasma yields approximately 2–10 ng of cfDNA, and we currently extract 1–5 mL of plasma from each sample. A standard library preparation can incorporate an input range of 1–50 ng cfDNA, so that in our case, the upper limit of input is often not reached. Extraction of 10–15 mL plasma would most often still enable one to input all of the extracted cfDNA and therefore enhance the lower limit of detection by a factor of 2–3.

Quantification of ctDNA is challenging, as minute amounts of mutated DNA fragments have to be detected in a large background of unmutated DNA. Especially in small, localized tumors, this fraction of ctDNA is often less than 0.01% of cfDNA, which is below the detection threshold of most assays [13,25]. Therefore, approaches which work independently of DNA mutations and monitor a global pattern of miRNAs in whole blood or a specific miRNA in serum were developed [29,30]. These seem to depend more on the mere presence of tumor cells, and less on tumor size, grading and stage. However, as these changes in blood are mainly indirectly related to the tumor, specificity is reduced compared to ctDNA breakpoint detection.

Beyond tumor surveillance, our assays and data have several implications in tumor biology and possible treatment prediction. Despite the characteristic translocation t(x;18)(p11.2;q11.2), point mutations were identified, such as in signal-transducing adaptor family member-2 (*STAP2*), which correlated with the clinical course and tumor burden. STAP2 is an adaptor protein and commonly promotes tumorigenesis in melanoma, CML, breast and prostate cancer [31,32,33]. Therefore, STAP2 potentially regulates signaling pathways, such as EGFR/PIK pathway, inducing sarcomas.

KIAA1328, STAP2 and MRPS16 ctDNA levels were high before the resection, declined post-op and increased again before the calvarial metastasis was detected. Another increase was observed with the detection of lung metastases. This points towards similar subclones from which the calvarial and lung metastases originated, as all metastases harbor a similar mutation profile.

In summary, targeted NGS and sequencing plasma DNA allow for highly sensitive and non-invasive detection of ctDNA in SS patients. Given our promising results, the methods we have described can be routinely applied clinically to detect tumor recurrence and monitor tumor heterogeneity. In addition, ctDNA levels could serve as a tool for risk stratification with regard to the indication for chemotherapy. They warrant closer investigation in prospective trials. The assay can easily be adapted to other translocation-driven tumors, which we have now displayed successfully in SS and MLS.

## 5. Conclusions

Here, we present a methodologically sound approach for ctDNA monitoring in synovial sarcoma patients. The disease- and patient-specific targeted NGS technique can be used in a routine diagnostic setup. Quantification of ctDNA can help to detect tumor recurrence and monitor treatment response with minimal invasiveness. These results warrant further investigation in larger patient cohorts so they can be translated from bench to bedside in a timely manner.

## Figures and Tables

**Figure 1 cancers-14-02078-f001:**
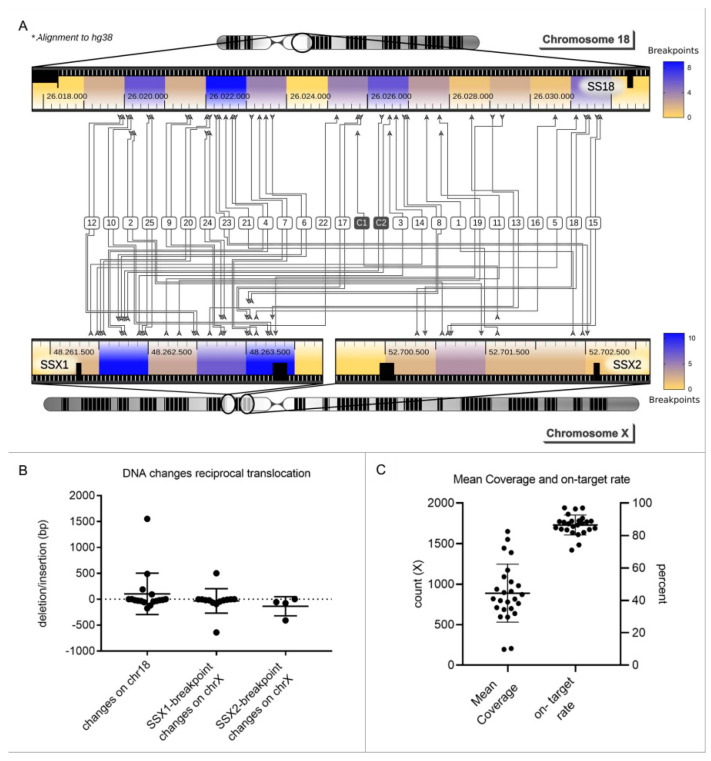
Mutational profiling of synovial sarcomas. (**A**). Heat map: 25 SS tumors and 2 cell lines (Fuji and HS-SY-II) were sequenced with the subtype-specific panel for SS. Chromosomal translocations could be detected in all 25 tumors (1–25) and both cell lines (C1, C2). Both breakpoints were found in 17/25 tumors (68%). Fusion occurred between *SS18* (*SYT*) in chromosome 18 and *SSX1* or *SSX2* in chromosome X. In total, 76% of the tumors had an *SSX1* breakpoint and the remaining 24% had a breakpoint in *SSX2*. Areas with an increased likelihood of chromosomal breaks are colored in blue. (**B**). For the 17 tumors where both breakpoints were identified, loss or gain of DNA during the translocation event could be determined. A mean change of 104 bp, standard deviation (SD) of 400 bp, occurred on chromosome 18. On chromosome X at *SSX1* and *SSX2*, a mean change of −32 bp (SD 235 bp) and −135 bp (SD 185 bp) occurred, respectively. Bars depict the mean and SD for 17 tumors. (**C**). Mean coverage of all tumors was 889 x (SD 357 x), and the mean on-target rate was 86.5% (SD 6.2%). Bars depict the mean and SD for 17 tumors.

**Figure 2 cancers-14-02078-f002:**
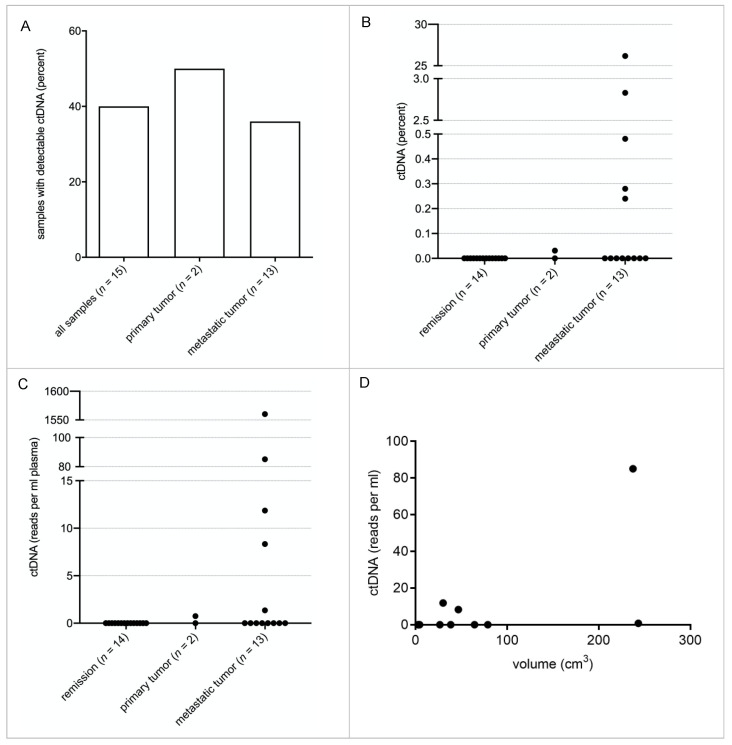
Quantification of ctDNA in SS patient plasma samples. A total of 29 plasma samples from 15 patients collected over 3 years of treatment and follow-ups were analyzed with the breakpoint panel derived from the data obtained with the subtype-specific panel. (**A**). ctDNA was detected in 40% of the total “active” samples (6 out of 15). One of two samples from primary tumor patients was positive, as well as 38.5% of those who had metastasis (5 out of 13 patients). None of the samples from patients in remission contained detectable ctDNA. (**B**). Depicted are the relative amounts of ctDNA for each plasma sample calculated as a fraction of cfDNA. (**C**). Depicted are the absolute amounts of ctDNA in reads/mL plasma for each plasma sample. (**D**). The absolute amounts of ctDNA are correlated to the detectable tumor volume. A correlation analysis shows a correlation coefficient of 0.55 with a *p*-value of 0.0276 (CN 95%) R^2^ coefficient of 0.563.

**Figure 3 cancers-14-02078-f003:**
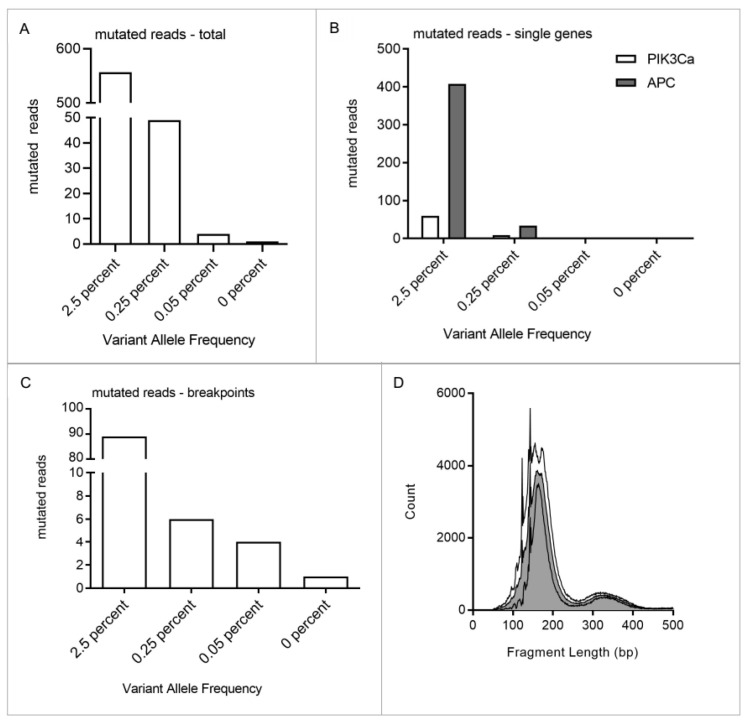
Dilution series of FUJI cell line DNA. DNA was fragmented and mixed with 10 ng cfDNA from a healthy donor to a VAF of 2.5%, 0.25%, 0.05% and 0%. Mutational profiling of the cell line and donor cfDNA revealed one breakpoint and mutations in *PIK3CA* and *APC*. A total of five mutations could be tracked, one breakpoint and four SNVs. (**A**). The lower limit of detection of the subtype-specific panel was determined by quantifying the mutated reads within each sample. Fuji cell line DNA was detectable even at a VAF of 0.05%. Respective values were: VAF 2.5%: 557 reads, VAF 0.25%: 49 reads, VAF 0.05%: 4 reads and VAF 0%: 1 read. (**B**). Mutated reads were further divided into breakpoint reads and reads with point mutations. Point mutations could be detected in *PIK3CA* and *APC* in 2.5% and 0.25% VAF samples. (**C**). Breakpoints could even be detected within the samples with the lowest VAF of 0.05%. One false positive event was identified in the 0% VAF sample. (**D**). Depicted are the fragment length distributions of each sample. Median fragment length is 182 bp.

**Figure 4 cancers-14-02078-f004:**
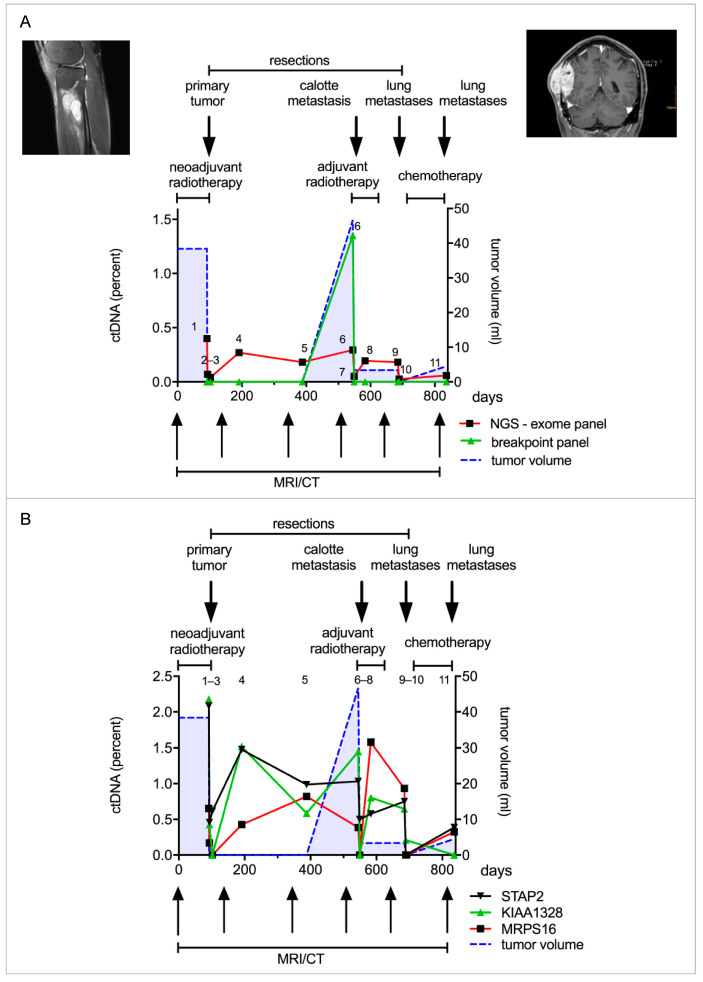
ctDNA quantification with a patient-specific panel derived from tumor exome sequencing. (**A**). Patient 6 was diagnosed with a synovial sarcoma in the right lower leg. He received neoadjuvant radiotherapy and the tumor was subsequently resected. He later developed a calvarial metastasis, which was resected, and additionally he developed metastases in both lungs, which were resected in parts. A patient-specific panel was designed from tumor exome sequencing to analyze the patient’s plasma samples. Eleven plasma samples were collected over the course of three years and analyzed with the patient-specific panel and breakpoint panel. The first sample analyzed (time point 1) was taken after radiotherapy, before resection of the primary tumor. Two plasma samples (sample 3 + 4) were taken in the two weeks following resection and a further eight samples in the following two years. ctDNA quantified by the patient-specific panel declined after resection of the primary tumor. An increase in ctDNA levels (sample 4) was detected one year prior to the detection of the calvarial metastasis. ctDNA again decreased after tumor resection and consequently increased with the occurrence of lung metastasis (red line). The breakpoint panel detected ctDNA only in the sixth time point just before resection of the calvarial metastasis (green line). The patient-specific panel is shown to be more sensitive than the breakpoint panel and is correlated with the clinical course. (**B**). Point mutations were identified in several genes, such as *STAP2*, *KIAA1328* and *MRPS16*. ctDNA declined following resection of the primary tumor and increased one year prior to detection of the calvarial metastasis.

## Data Availability

The data presented in this study are available on request from the corresponding author. The data are not publicly available due to privacy restrictions.

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
