# Peer review of "Genotyping of Circulating Free DNA Enables Monitoring of Tumor Dynamics in Synovial Sarcomas"

_cancers, 2022, doi:10.3390/cancers14092078_

Round 1

Reviewer 1 Report

This is a very interesting investigation on liquid biopsy in synovial sarcoma, notably it is one of the first.

First of all, I would like to congratulate the authors for their highly important results. 

However, some minor changes might enhance the significance of their work:

Nowadays the mortality and the metastatic rate of SS is far lower than mentioned and cited. The references are relatively old, and unfortunately, every SS expert will see that at once. Read: Venkatramani, R., et al., Synovial Sarcoma in Children, Adolescents, and Young Adults: A Report From the Children's Oncology Group ARST0332 Study. J Clin Oncol, 2021. 39(35): p. 3927-3937 and Scheer, M., et al., The effect of adjuvant therapies on long-term outcome for primary resected synovial sarcoma in a series of mainly children and adolescents. J Cancer Res Clin Oncol, 2021 to get the current outcomes. While reported outcomes differ between pediatric and adult series, the outcomes reported in a very general matter may be only true for SS patients with advanced disease/ high risk patients - with the current treatment approaches.

Follow-up care for SS is not difficult and not costly. Compared to other sarcoma diseases it is very simple.  Metastases are commonly in the lungs. Extra-lung metastases are an absolute exception in SS (while they may be more often in other sarcoma diseases). And local relapse may be detected with ultrasound even though we prefer MRI. Unfortunately, recent results for rhabdomyosarcoma did not reveal a benefit of surveillance imaging: Fetzko, Stephanie et al. 2022: Is Detection of Relapse by Surveillance Imaging asociated with longer Survival in Patients with Rhabdomyosarcoma? J Pediatr Hematol Oncol doi: 10.1097/MPH.0000000000002429

As it can not be excluded that other groups might have similar results, I would be careful and include in the discussions a small comment about: Is it true that early tumor detection is associated with a survival benefit? (If not what role might liquid biopsy play in the future? Or might it add additional information?)

I think it is very interesting, that larger SS tumors tend to shed more ctDNA. Do you think that ctDNA might be used as an additional factor for risk stratification of SS? Especially with regard to the question of the indication for chemotherapy/ tumor aggressiveness?

Reviewer 2 Report

The study conducted by Anja Eisenhardt et al. first applied a subtype-specific targeted NGS panel for breakpoint detection in FFPE tissue from 25 synovial patients, yielding high sensitivity and specificity of 100%. Then, a breakpoint lockdown panel specific to each patient was performed on 29 plasma samples from 15 patients, which, despite a high specificity of 100 %, only exhibited a limited sensitivity to 40%. As such, a patient-specific panel based on exome sequencing for one patient with metastatic disease was developed as a boost, targeting on both breakpoint region and point mutations. The ctDNA level was highly correlated with the clinical course, response to treatment and tumor volume. This manuscript is well written and the results shed light on the potential use of circulating tumor DNA as a noninvasive method for the monitoring of tumor dynamics. I have only some minor comments or questions as follows:

Q1. Previous studies investigating the potential of liquid biopsy in synovial sarcoma by Mihály et al. (PMID: 30326929), Przybyl et al. (PMID: 30871572), and authors’ team (PMID: 27069481), had shown that circulating SYT-SSX fusion gene was an infrequent event and did not constitute a sufficiently reliable biomarker to monitor tumor dynamics. Detection of ctDNA by breakpoint sequencing in the current study revealed a similar low sensitivity of 40%. Leaving aside the driver gene fusion of SYT-SSX, SSs are known to be largely quiet in mutations. Please explain to what extent the assay sensitivity can be improved by incorporating point mutations and what other strategy the authors think of can further increase the sensitivity of liquid biopsy for SS patients in the discussion.

Q2. Recent studies have reported the promising role of blood-borne miRNA signature   in synovial sarcoma as a biomarker for diagnosis and monitoring of local recurrence or distant metastasis by authors’ team (Fricke et al. PMID: 26250552) and Uotani et al. (PMID: 29116117). In the discussion, the authors should compare pros and cons of in miRNA signature vs. the mutations detected by exome sequencing.   

Q3. It appears costly to perform molecular analysis for liquid biopsy in all synovial sarcoma patients. The authors need to explain how the assay could be integrated into a feasible clinical workflow in real world.

Reviewer 3 Report

The authors clearly present clinical diagnostic assays for the detection of synovial sarcomas from circulating tumor DNA. The authors state that this approach is based upon similar analyses done to detect other tumor types. They are transparent about the benefits and limitations of this tool and show exciting clinical followup from one patient. Obviously having this data from multiple patients would be ideal. Nevertheless, this is an important clinical study which would be of great importance to other orthopedic oncologists and sarcoma scientists. 

Author Response

We thank the referee for critically reviewing our manuscript, and are happy that the overall impression is positive. We are currently planning a larger trial to investigate ctDNA in a larger patient cohort and are confident that we can soon provide data from multiple patients for in depth analysis of ctDNA in Synovial Sarcomas.